# IL-6 Enhances the Negative Impact of Cortisol on Cognition among Community-Dwelling Older People without Dementia

**DOI:** 10.3390/healthcare11070951

**Published:** 2023-03-25

**Authors:** Eirini Koutentaki, Maria Basta, Despina Antypa, Ioannis Zaganas, Symeon Panagiotakis, Panagiotis Simos, Alexandros N. Vgontzas

**Affiliations:** 1Department of Psychiatry and Behavioral Sciences, School of Medicine, University of Crete, Voutes, Heraklion, 71003 Crete, Greece; 2Department of Psychiatry, University Hospital of Heraklion, Voutes—Heraklion, 71110 Crete, Greece; 3Sleep Research and Treatment Center, Department of Psychiatry, Penn State University, Hershey, PA 17033, USA; 4Department of Neurology, School of Medicine, University of Crete, Voutes, Heraklion, 71003 Crete, Greece; 5Department of Internal Medicine, University Hospital of Heraklion, Panepistimiou Ave., Heraklion, 71500 Crete, Greece; 6Computational Biomedicine Lab, Institute of Computer Science, Foundation for Research and Technology (FORTH), 100 Nikolaou Plastira Str., Vassilika Vouton, Heraklion, 70013 Crete, Greece

**Keywords:** basal cortisol, inflammatory markers, older people, executive function, MCI

## Abstract

There is growing evidence that high basal cortisol levels and systemic inflammation independently contribute to cognitive decline among older people without dementia. The present cross-sectional study examined (a) the potential synergistic effect of cortisol levels and systemic inflammation on executive function and (b) whether this effect is more prominent among older people with mild cognitive impairment (MCI). A sub-sample of 99 patients with MCI and 84 older people without cognitive impairment (CNI) (aged 73.8 ± 7.0 years) were recruited from a large population-based cohort in Crete, Greece, and underwent comprehensive neuropsychiatric and neuropsychological evaluation and a single morning measurement of cortisol and IL-6 plasma levels. Using moderated regression models, we found that the relation between cortisol and executive function in the total sample was moderated by IL-6 levels (*b* = −0.994, *p* = 0.044) and diagnostic group separately (*b* = −0.632, *p* < 0.001). Moreover, the interaction between cortisol and IL-6 levels was significant only among persons with MCI (*b* = −0.562, *p* < 0.001). The synergistic effect of stress hormones and systemic inflammation on cognitive status appears to be stronger among older people who already display signs of cognitive decline. Targeting hypercortisolemia and inflammation may be a promising strategy toward improving the course of cognitive decline.

## 1. Introduction

Aging in the global population is associated with an increased burden of cognitive impairment, including for dementia [1]. The detection of factors associated with cognitive dysfunction is a field of increased interest both for the treatment of vulnerable individuals and the implementation of strategies to delay age-related cognitive decline [2,3].

It is known that high basal cortisol levels and systemic inflammation independently contribute to cognitive decline among older people without dementia. Specifically, glucocorticoids are strongly linked to memory performance [4,5,6,7]. Additionally, hypothalamic–pituitary–adrenal axis (HPA axis) dysregulation may be linked to stress-related factors and is associated with cognitive impairment among older adults [8,9,10]. Many studies have suggested that elevated cortisol levels inhibit neurogenesis, lead to reduced neocortical and hippocampal volume, and have a negative impact on memory function [11,12]. Moreover, other studies in older people have revealed negative associations between basal cortisol levels and cognitive performance [13], while elevations in cortisol appear to be a risk factor for greater cognitive decline in global cognition, verbal memory, and executive functioning [14,15,16,17] in this age group with or without cognitive impairment. Finally, we recently reported that cortisol is elevated in patients with MCI and Alzheimer’s disease (AD) compared to cognitively non-impaired controls [17,18].

Recently, studies have shown that brain pathology in dementias may be related to the activation of the inflammatory pathways [19,20,21]

Moreover, pro-inflammatory cytokine levels (such as IL-6 and TNFα) have been found to be elevated in patients with dementia [22,23,24,25] However, many reports on cytokine levels in MCI or AD patients are controversial or inconclusive, particularly those which provide data on frequently investigated cytokines such as tumor necrosis factor alpha (TNF-α) or interleukin-6 (IL-6) [26,27,28,29]. In view of evidence that cytokine (e.g., IL-6) levels appear to be inversely correlated with hippocampal volume [30,31], it has been hypothesized that IL-6 may potentiate memory impairment, playing a possible pathogenic role in Alzheimer’s disease (AD), vascular dementia, and age-related cognitive decline [32,33].

Stress hormones and pro-inflammatory cytokines appear to relate to each other in multiple direct and indirect ways [34]. Cortisol seems to inhibit both the transcription and actions of pro-inflammatory cytokines [35,36]. On the other hand, cytokines lead to increased cortisol levels, either directly by inducing cortisol release [37,38], or indirectly by inhibiting the glucosteroid receptor translocation and signaling [39,40,41].

However, the literature examining the combined effect of high basal cortisol levels and inflammation is limited. In a previous pilot study conducted in cognitively intact older people, the authors showed that higher cortisol, in conjunction with higher IL-6 and TNF-alpha, is associated with smaller hippocampal volume [42], an area critical for cognition, and long-term cognitive decline if structure or function is disturbed [43,44,45]. However, up to now, to our knowledge, no studies have examined the synergistic effect of both basal stress hormone and inflammation levels on neurocognitive indices, such as performance on executive function tests, in older people either with or without cognitive impairment.

To fill the gap in the existing literature, the aim of this study was to examine (a) the potential synergistic effect of cortisol levels and systemic inflammation on the executive function of community-dwelling older people without dementia and (b) whether this effect is more prominent among patients with mild cognitive impairment (MCI) compared to older people who are cognitively intact. We hypothesized that higher chronic stress levels, measured by basal plasma cortisol levels, combined with higher inflammation may inversely relate to executive function among older people and this effect may be more prominent in those with mild cognitive impairment (MCI) compared to older people without cognitive impairment (CNI).

## 2. Materials and Methods

### 2.1. Study Design

The current sample consisted of participants in the Cretan Aging Cohort, a cross-sectional study of community-dwelling older people, recruited from the district of Heraklion, Crete, Greece. The study was conducted between March 2013 and June 2015, and its primary aim was to investigate the prevalence of and risk factors associated with cognitive decline [46]. The Cretan Aging Cohort study was conducted in two phases (Figure 1), as described in detail in our previous publications [46,47], according to the guidelines of the Declaration of Helsinki. All procedures including human participants were approved by the Bioethics Committee of the University Hospital of Heraklion, Crete, Greece (Protocol Number: 13541, 20-11-2010). Additionally, written informed consent was obtained from all participants.

Phase I: Eligible participants were those aged >60 years who visited selected primary health care facilities in areas of the Heraklion district for any reason. All consenting individuals (*n* = 3200) completed an interview with a trained nurse, using a structured questionnaire to assess sociodemographic information, anthropometric measurements, and physical and mental health, as well as medication use and lifestyle. Cognitive function was evaluated using the Greek version of the Mini Mental State Examination (MMSE) [48], applying a universal cut-off of 23/24 points (since the majority of participants had ≤6 years of formal education) for referral of patients for further evaluation. Based on this cut-off, participants were divided into two groups: those with MMSE < 24, considered to be at risk for cognitive impairment, and the not-at-risk group with MMSE ≥ 24 [46]. After excluding participants with crucial missing data (MMSE score, age), the final study sample consisted of 3140 people (57.0% women) aged 73.7 ± 7.8 (60–100) years, who had completed an average of 5.8 ± 3.3 (0–18) years of formal education and lived mostly in rural areas.

Phase II: Participants who scored <24 points on the MMSE (*n* = 636) were invited for an extensive neuropsychological and neuropsychiatric evaluation in phase II of the study. Among them, 344 older people agreed to participate (response rate 54%). Consenting participants did not differ from the 636 originally invited in terms of age, gender, and body mass index (BMI). Certified neurologists, psychiatrists, and internists completed an extensive questionnaire modified from the one employed in the HELIAD study [49]. Neuropsychological assessment was performed by trained neuropsychologists based on a test battery evaluating a variety of cognitive domains. Medical and family history were also assessed. Diagnosis of any type of MCI was based on modified Petersen criteria (IWG-1) [50] and on a consensus decision between two or more clinicians who took into account results from the comprehensive neuropsychiatric and neuropsychological evaluation. Diagnosis of MCI further required that cognitive deficits could not be accounted for by clinically significant mood or anxiety disorder.

To be included in the MCI group, participants had to have age- and education-adjusted z scores < −1.5 on indices derived from at least two tests within a given cognitive domain (episodic memory, language, attention/executive) and demonstrate intact levels of everyday functionality according to a comprehensive, informant scale of instrumental activities of daily living [51] adapted for the Greek population from Lawton and Brody (1969). Using the non-cognitively impaired pool of subjects who scored >24 on the MMSE (*n* = 2504), a control group of 181 participants was created after stratifying for residence, gender, and age. Of these, 161 agreed to participate in phase II of the study (see Figure 1). In phase II, among the 505 participants examined, 231 were diagnosed with MCI, 128 with dementia, and 146 were CNI [46].

### 2.2. Participants

The final subsample included in this sub-analysis consisted of 183 participants, 99 with MCI and 84 CNI (124 women and 59 men), aged between 60 and 92 years, with 0–17 years of education, BMI between 20.2 and 45.94 kg/cm^2^, and waist circumference between 72 and 133 cm (Table 1). Among the MCI participants, 28 (28.3%) presented with neurocognitive profiles consistent with the pure amnestic, 41 (41.4%) with the amnestic–multidomain, and 30 (30.3%) with the non-amnestic subtypes.

### 2.3. Measurements

#### 2.3.1. Cortisol and Inflammatory Markers

As previously described [18,25], single morning blood samples were collected between 10:00 a.m. and 12:00 p.m., transferred to EDTA-containing tubes (3 per patient), and refrigerated until centrifugation (within 3 h) for plasma isolation, which was kept in deep freeze (−80 °C). Plasma cortisol was measured using the ELISA technique (Cusabio Human cortisol ELISA kit, cfb-e05111h, Cusabio Technology LLD, Houston, Texas, USA). The inter- and intra-assay coefficients of variation were 23.1% and 16.1%, respectively. The lower detection limits for cortisol were 0.02 ngr/mL.

Plasma IL-6 was measured using the ELISA technique (Human IL-6 Quantikine HS ELISA kits, R&D Systems Europe, Abington, UK). For the IL-6 determination, the inter-assay coefficients of variation were 13.9%, the intra-assay coefficients of variation were 11.04 and the lower detection limits was 0.133 pg/mL.

#### 2.3.2. Executive Function Tests

In view of the diversity of executive functions, several tests were administered to each participant and individual scores were averaged to provide a single executive index [51,52].

(a)Digit Reverse subtest from the Greek Memory Scale [53]. This subtest is considered a working memory index. The task requires the repetition of single digit sequences in reverse order, with seven difficulty levels, ranging from 2 to 8 digits. Successful repetition of all digits without additions and in correct reverse order is scored with two points, while successful repetition of all digits in correct reverse order with only one switch in the correct (reverse) order of two digits is scored with one point, with a maximum of 24 points.(b)Semantic Verbal Fluency test (SVF; [54]). In this task, the participant has to name as many words as possible that begin with a given letter within 60 s. The participant should not give words with the same root but different endings, proper names, or numbers.(c)Symbol Digit Modalities test assessing visuomotor processing speed and sustained attention (SDMT; [55]).(d)Trail Making Test Part B (TMT-B assessing set-shifting ability; [56]). The task requires the participants to trace a line that connects circled numbers and circled letters in consecutive order while alternating between numbers and letters (e.g., 1-A-2-B-3-C).

In addition, the Greek research adaptation of PPVT-R was administered to measure cognitive reserve (lexical knowledge and processing ability) not accounted for by years of formal education. Raw scores from each test were converted to age- and education-adjusted standard (z) scores, using the regression method proposed by Petersen et al. (1992) [57], based on Greek normative data reported elsewhere [58]. This transformation permitted the computation of domain composite scores through the averaging of individual raw scores on relevant tests.

#### 2.3.3. Emotional Status

Self-reported depression symptoms were assessed using the Greek version of the Geriatric Depression Scale (GDS; [48]), which has been translated and validated in Greek. The Geriatric Depression Scale-15 (GDS-15) is a short, 15-item instrument specifically designed to assess depression in geriatric populations. Its items require a yes/no response.

### 2.4. Statistical Analysis

The first hypothesis of the present study was assessed with a moderated mediation model with the log-transformed IL-6 levels as a potential moderator of the relation of the log-transformed basal cortisol levels with the executive function performance index (as described above), adjusted for sex, waist circumference, depression diagnosis, and performance in a receptive vocabulary task, as a representative measure of educational level (Figure 2).

The second hypothesis of the study was assessed with a moderated mediation model with the log-transformed IL-6 levels and diagnostic group (NI vs. MCI) as potential moderators of the relation of the log-transformed basal cortisol levels with the executive function performance index, adjusted for sex, waist circumference, depression diagnosis, and performance in a receptive vocabulary task, as a representative measure of educational level (Figure 3). The alpha level was set to 0.05 for all statistical tests. All analyses were performed with IBM SPSS Statistics, Version 25 (SPSS Inc., Chicago, IL, USA); mediation and moderated mediation analyses were performed with the use of the PROCESS macro (processmacro.org; [59]). The conditional effects of the independent on the dependent variable in Process are estimated by default at three levels of the moderator variable (IL6), which correspond to the 25th (0.64 pg/mL), 50th (1.05 pg/mL), and 75th percentiles (2.07 pg/mL).

## 3. Results

### 3.1. Is the Relation of Basal Cortisol Levels with Executive Function Moderated by IL-6 Levels?

The relation of basal cortisol levels with executive function was moderated by IL-6 levels in the total sample [*b* = −0.830, 95% CI [−1.505, −0.155], *t*(171) = −2.428, *p* = 0.016]. This effect was only significant among participants with higher IL-6 levels [*b* = −0.229, 95% CI [−0.599, −0.120], *t*(171) = −2.960, *p* = 0.004], not those with moderate or lower IL-6 levels (all *p* > 0.138). Additionally, the regression coefficients between cortisol levels and executive function [*b* = −0.169, 95% CI [−0.330, −0.009], *t*(171) = −2.078, *p* = 0.039], as well as between IL-6 levels and executive function, were statistically significant [*b* = −0.571, 95% CI [−1.105, −0.038], *t*(171) = −2.115, *p* = 0.036]. Among the covariates, only the performance in the receptive vocabulary task was statistically significant [*b* = 0.037, 95% CI [0.019, 0.054], *t*(171) = 4.069, *p* < 0.001].

### 3.2. Is the Moderation by IL-6 Levels of the Relation of Basal Cortisol Levels with Executive Function Moderated by Diagnostic Group (CNI vs. MCI)?

In the total sample, the relation of basal cortisol levels with executive function was moderated by IL-6 levels [*b* = −0.994, 95% CI [−1.961, −0.027], *t*(167) = −2.029, *p* = 0.044] and diagnostic group separately [*b* = −0.632 s, 95% CI [−0.901, −0.363], *t*(167) = −4.643, *p* < 0.001]. Moreover, there was an interaction of cortisol levels and diagnostic group with executive function [*b* = −0.562, 95% CI [−0.872, −0.253], *t*(167) = −3.590, *p* < 0.001], but no other interactions between IL-6 levels and diagnostic group, and cortisol and IL-6 levels and diagnostic group with executive function performance (all *p* > 0.644).

In the sub-group of CNI participants, there were no statistically significant effects on or interaction of cortisol and IL-6 levels with executive function (all *p* < 0.157). In the sub-group of MCI participants, however, the interaction between cortisol levels and executive function was statistically significant [*b* = −0.420, 95% CI [−0.640, −0.200], *t*(88) = −3.795, *p* < 0.001]. There were no other statistically significant effects or interactions (all *p* > 108).

## 4. Discussion

The main findings of this study are that higher plasma cortisol levels combined with higher inflammation inversely relate to executive function among older people, and this effect appears be more prominent in those with MCI compared to older people without cognitive impairment.

The first key finding of this study is the synergist negative effect of basal cortisol levels (likely reflecting the accumulated impact of chronic stress) and pro-inflammatory cytokines on executive function in older people. It is known that cortisol and cytokines interact with each other with various mechanisms. Stress not only activates the sympathetic nervous system and HPA axis, but also components of the immune system. Thus, proinflammatory cytokine production increases in response to acute psychological stress in humans [60], and chronic stress can elevate the plasma levels of inflammatory cytokines [61]. Conversely, cortisol release can be upregulated by circulating pro-inflammatory cytokines, while IL-1Beta and IL-6 can activate the HPA axis [62,63].

Previous studies suggest that the chronic increase in cortisol levels is associated with cerebral atrophy [64]. Recently, it was concluded that cortisol and cytokines interact and affect hippocampal volume and physiology in the older population without cognitive impairment [13]. Cortisol inhibits neurogenesis and increases the risk of cell death [65]. Moreover, the chronic activation of the HPA axis and high cortisol levels can lead to hippocampal atrophy neuronal damage, neuroinflammation, and cognitive impairment [31,64]. IL-6 has a negative effect on hippocampal gray matter volumes [16], and IL-1 and TNF-Alpha have also been shown to inhibit synaptic plasticity in the hippocampus and contribute to neurotoxicity [35,66,67]. Reported hippocampal atrophy was based on MRI hippocampal volume measurements, while none of these studies focused on the combined effect of stress and inflammation on executive function in cognitively impaired and non-impaired populations.

In our study, we have focused on the combined effect of cortisol and cytokines on executive function in older adults, providing evidence that the accumulated synergistic action of circulating IL-6 and cortisol may negatively impact executive functions. It is known that chronic stress triggers an immune response in older adults [68,69], while on the other hand, studies have shown that old age is associated with increased IL-6 secretion triggering the elevation of cortisol levels [70]. A recent study in older people without cognitive impairment has shown that high subjective perceived stress at baseline accelerated age-related elevations of cytokines and decline in executive capacity over time. Importantly, these monocyte/macrophage cytokines partially mediated the relationship between age and executive function in this group [71]. However, this study did not include older people with cognitive impairment and, moreover, stress levels were subjectively assessed and measured via the Perceived Stress Scale rather than using biomarkers.

The second significant finding of the present study is that the synergistic effect of cortisol with inflammatory levels is evident in older participants with mild cognitive impairment but not in older people without cognitive impairment. Thus, persons who already display signs of neurocognitive decline appear to be more vulnerable to the combined effect of hypercortisolemia with elevated inflammation, compared to cognitively intact older individuals. To our knowledge, there are no studies examining the possible differential effect of stress and inflammation levels on executive function in both MCI and CNI older adults. Previous studies have documented the association of cortisol [8,9,17,18] or cytokines [19,22,25] with cognitive impairment separately. Thus, the current results complement the findings of Casaletto et al. [71] by including an objective measure of accumulated, chronic stress (i.e., basal cortisol) and extend them to older people with MCI. We suppose that in patients with mild cognitive impairment, there are higher levels of cortisol and cytokines due to more emotional stress and underlying mechanisms that increase inflammation compared to controls.

The increased vulnerability of the MCI patients compared to the CNI group could be associated with the pre-existing neuroinflammation and the aging of the immune system [72,73], as well as with the chronic stress and the higher levels of cortisol in these patients [74,75] compared to those without cognitive impairment. This confirms the relationship between stress, inflammation, and cognitive impairment in older adults.

### 4.1. Strengths and Limitations

This study has significant strengths. Firstly, our study included a relatively large naturalistic sample of community-dwelling older people without dementia in a well-defined population assessed with an extensive neuropsychiatric and neuropsychological evaluation. Furthermore, no exclusion criteria were applied in recruitment, leading to a final sample representative of the older adult population in a rural southern European region. In addition, the associations examined were controlled for emotional variables (i.e., self-reported depression symptoms based on the Greek version of the Geriatric Depression Scale).

However, our study also has certain limitations. The cross-sectional design of the study does not permit the evaluation of causal effects between basal cortisol levels, inflammation markers, and executive function. Additionally, cortisol and inflammation levels were not assessed across 24 h, but were based on a single morning measurement. Furthermore, no structural or functional imaging was included. Therefore, no additional information about the impact of cortisol levels on brain structures, such as the hippocampus, could be provided. Finally, the sample size did not allow analysis stratified by MCI sub-groups (amnestic, non-amnestic, etc.).

### 4.2. Implications

Our findings have several clinical implications. Hypercortisolemia appears to be a risk factor for more severe cognitive deficits in older people with MCI. Modifiable factors associated with increased cortisol levels, such as depression, anxiety, and sleep disturbances in this population, should be thoroughly screened for and treated. Hyperactivity of the HPA axis could be reversed by anti-depressant therapy [76,77] and, moreover, the development of anxiety and mood disorders could be attenuated by medication with anti-anxiety effects, such as benzodiazepines, and selective serotonin reuptake inhibitors or other antidepressants [78,79]. Additionally, since inflammation has a synergistic effect on cognition, a possible treatment approach could be the use of anti-inflammatory agents for patients with MCI, which may help reverse or delay cognitive decline. In this context, there is preliminary evidence of a positive effect of ibuprofen on hippocampal neurogenesis and memory [80,81]. However, more studies are needed to examine the possible neuroprotective effect of non-steroidal anti-inflammatory drugs, as they may attenuate the effects of the modulators of inflammation that have been implicated in the pathogenesis of cognitive impairment.

## 5. Conclusions

In sum, higher chronic stress, as reflected in higher basal plasma cortisol levels, and higher systemic inflammation, as expressed by IL-6 levels, are inversely related to executive capacity among older people without dementia, and this effect is more prominent among patients with mild cognitive disorder (MCI) compared to older people without cognitive impairment (CNI). Treating conditions associated with hypercortisolemia (such as depression, anxiety, and insomnia) and targeting inflammation may be a promising strategy toward improving/delaying the course of cognitive decline. Future longitudinal studies may further elucidate the complex relations between cortisol, inflammation, and cognition, as well as the effect of therapeutic interventions targeting stress and inflammation in the course of cognitive function in older adults with cognitive deficits.

## Figures and Tables

**Figure 1 healthcare-11-00951-f001:**
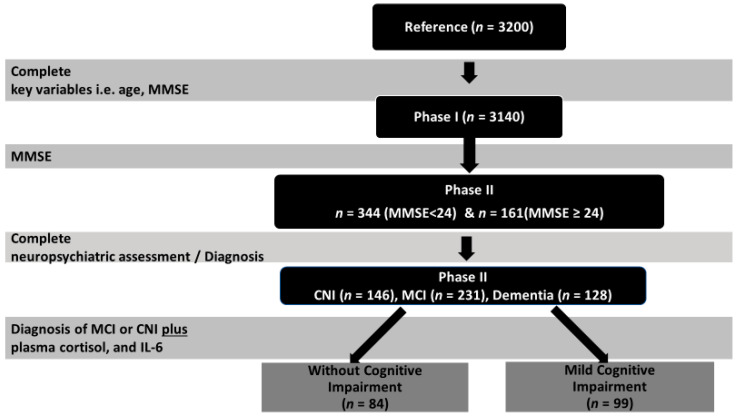
Study flowchart. MCI, mild cognitive impairment; CNI, without cognitive impairment; MMSE, Mini Mental State Examination.

**Figure 2 healthcare-11-00951-f002:**
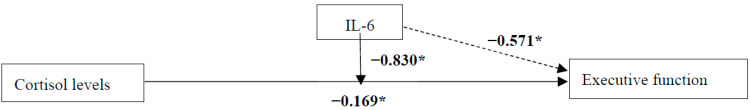
Unstandardized regression coefficients for the relationship between cortisol levels and executive function with moderator IL-6 in the total sample. Sex, waist circumference, depression diagnosis, and performance in a receptive vocabulary task, as a measure of educational level, were included as confounders in all models. * *p* ≤ 0.05.

**Figure 3 healthcare-11-00951-f003:**
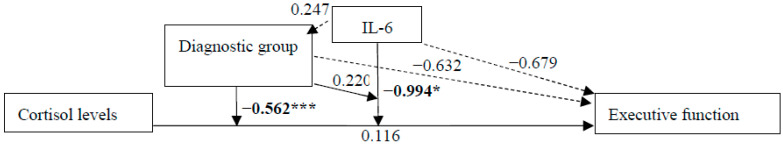
Unstandardized regression coefficients for the relationship between cortisol levels and executive function with moderators IL-6 and diagnostic group (CNI vs. MCI) in the total sample. Sex, waist circumference, depression diagnosis, and performance in a receptive vocabulary task, as a measure of educational level, were included as confounders in all models. * *p* ≤ 0.05; *** *p* ≤ 0.001.

**Table 1 healthcare-11-00951-t001:** Sample characteristics.

	Total Sample	CNI Group	MCI Group	*p* Value
N	183	84	99	
Age (years)	73.8 ± 7.0	72.7 ± 7.4	75.3 ± 6.4	0.01
Education (years)	5.37 ± 3.1	5.82 ± 2.93	4.83 ± 3.25	0.2
Gender (%)				0.2
Women	67.7	65.5	74.7	
Men	32.2	34.5	25.3	
BMI (kg/m^2^)	30.0 ± 4.79	30.32 ± 5.06	29.59 ± 4.44	0.4
Waist circumference (cm)	102.8 ± 3.5	103.4 ± 13.2	102.3 ± 11.3	0.6
Marital status (%)				0.3
Single/divorced	5.0	3.6	7.1	
Married	73.2	77.1	67.7	
Widowed	21.8	21.4	25.3	
GDS	3.71 ± 3.50	3.22 ± 3.46	4.12 ± 3.53	0.09
MMSE score	24.72 ± 3.56	26.89 ± 2.90	22.89 ± 3.00	<0.001
Cortisol (ng/mL)	76.4 ± 123.6	60.5 ± 137.6	72.7 ± 118.2	0.024
IL6 (pg/mL)	1.31 ± 0.81	1.28 ± 0.91	1.34 ± 0.71	0.4

Values depict means ± standard deviation, unless otherwise specified.

## Data Availability

Data available on request due to restrictions (privacy). The data presented in this study are available on request from the corresponding author. The data are not publicly available due to privacy restrictions.

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
