# Peer review of "IL-6 Enhances the Negative Impact of Cortisol on Cognition among Community-Dwelling Older People without Dementia"

_healthcare, 2023, doi:10.3390/healthcare11070951_

Round 1

Reviewer 1 Report

GENERAL COMMENTS

  Although IL-6 and cortisol are expected to interact in multiple ways (as explicitly acknowledged in the Introduction), such interactions are not shown in the diagrams.  Are IL-6 and cortisol associated in the study group? If there was an association, would it affect the argument of synergistic effect? For example, if it turned out that IL-6 and cortisol are positively associated, would it be valid to argue that IL-6 works, at least partially, as a mediator of cortisol effect on executive function? Please discuss.

  If I understand correctly the relevant sentence in the Results (Lines 211-212), performance in a receptive vocabulary task (representative measure of educational level) is strongly associated with executive function.
If this is correct, it raises certain questions: Is educational level associated with IL-6 and / or with cortisol? If so, a plausible hypothesis for the findings of this and other studies would be that IL-6 and cortisol are not independent or independently synergistic determinants of executive function but mediators of an education effect. Please discuss.

SPECIFIC COMMENTS

Results, Line 206

  b=-.229 instead of b=.229

Author Response

Reviewer 1

GENERAL COMMENTS

  Question 1. Although IL-6 and cortisol are expected to interact in multiple ways (as explicitly acknowledged in the Introduction), such interactions are not shown in the diagrams.  Are IL-6 and cortisol associated in the study group? If there was an association, would it affect the argument of synergistic effect? For example, if it turned out that IL-6 and cortisol are positively associated, would it be valid to argue that IL-6 works, at least partially, as a mediator of cortisol effect on executive function? Please discuss.

Answer: Thank you for this comment that raises an important issue in interpreting the current results. Please note that the two measures are weakly correlated in the total sample (r=0.08, p=0.27), making the hypothesis of mediation implausible.

  If I understand correctly the relevant sentence in the Results (Lines 211-212), performance in a receptive vocabulary task (representative measure of educational level) is strongly associated with executive function. If this is correct, it raises certain questions: Is educational level associated with IL-6 and / or with cortisol? If so, a plausible hypothesis for the findings of this and other studies would be that IL-6 and cortisol are not independent or independently synergistic determinants of executive function but mediators of an education effect. Please discuss.

Answer: We apologize for the potential confusion that this section of the results may have caused. Indeed, performance on the PPVT, which requires selection of one out of four line drawings on a page that best matches the meaning of a spoken word was moderately correlated with the executive index (r=0.472). This is similar to what we have reported previously on a large normative Greek sample aged 50-84 years (Simos, Kasselimis, Mouzaki, 2011), and earlier work demonstrating similar correlations between PPVT scores and performance IQ estimates using tests that place significant demands upon executive functions (Stevenson, 1986). In the current sample education level did not correlate with IL6 (r=0.019) or cortisol (r=-0.059). Therefore it is probably safe to surmise that the significant effect of PPVT-R as a covariate in the model was mainly due to the its common variance with the executive index.

Simos, P.G., Kasselimis, D., & Mouzaki, A. (2011). Age, gender, and education effects on vocabulary measures in Greek. Aphasiology, 25, 492-504.

Stevenson JD Jr. Alternate form reliability and concurrent validity of the PPVT-R for referred rehabilitation agency adults. J Clin Psychol. 1986, 42(4):650-3.

SPECIFIC COMMENTS

Results, Line 206

  b=-.229 instead of b=.229

Answer: We apologize for the type-o which has been corrected.

Reviewer 2 Report

Dear authors, following are only small comments for the revision of the manuscript. The manuscript is interesting and clearly described.

General: Please avoid the words elderly and demented, dementias,... in the manuscript. They are considered discriminatory in the international literature. Use e.g. older people and older people without dementia!

Abstract:
Please include your type of research design.

Indroduction:
Line 47: AD - please do not abbreviate the first time you use it.

Author Response

Reviewer 2

Dear authors, following are only small comments for the revision of the manuscript. The manuscript is interesting and clearly described.

We thank the reviewer for the positive comments.

1.Please avoid the words elderly and demented, dementias,... in the manuscript. They are considered discriminatory in the international literature. Use e.g. older people and older people without dementia!
Answer: Based on the reviewer’s comment we have now replaced elderly and demented with older people with or without dementia.

  1. Abstract:
    Please include your type of research design.

Answer: Based on the suggestion “cross sectional” has now been added in the abstract reflecting the research design

3. Introduction:
Line 47: AD - please do not abbreviate the first time you use it.

Answer: AD has now been replaced by “Alzheimer’s Disease”

Reviewer 3 Report

This manuscript aims to examine the potential synergistic effect of cortisol levels and systemic inflammation on executive function and whether this effect is more prominent among elderly with Mild Cognitive Impairment (MCI). The subject is very interesting and the manuscript is well structured. I have the following concerns:

- Line 110, The total number of participants with MMSE scores <24 was 636, while the total number of participants with MMSE scores <24 in Figure 1 was 344. Which one is correct?

- Line 206-207, What are the criteria for high, moderate, or low IL-6 levels? Many of the results and conclusions are based on the criteria, and a clear basis for the classification needs to be given.

- Why is it important to use multiple methods to test the executive function of subjects and which test result was used in the analysis?

- It is highly recommended to modify Figure 2 and Figure 3, as the images are not of high quality.

Author Response

Reviewer 3

This manuscript aims to examine the potential synergistic effect of cortisol levels and systemic inflammation on executive function and whether this effect is more prominent among elderly with Mild Cognitive Impairment (MCI). The subject is very interesting and the manuscript is well structured. 

Answer: We thank the reviewer for your positive comments.

  1. Line 110, The total number of participants with MMSE scores <24 was 636, while the total number of participants with MMSE scores <24 in Figure 1 was 344. Which one is correct?

Answer: We apologize to the reviewer for the confusion

  1. Line 206-207, What are the criteria for high, moderate, or low IL-6 levels? Many of the results and conclusions are based on the criteria, and a clear basis for the classification needs to be given.

Answer: The conditional effects of the independent on the dependent variable in Process are estimated by default at three levels of the moderator variable (IL6) which correspond to the 25th (0.64 pg/mL), 50th  (1.05 pg/mL) and 75th percentiles (2.07 pg/mL). This is now added in the revised text, in Results section.

  1. Why is it important to use multiple methods to test the executive function of subjects and which test result was used in the analysis?

Answer: It is well documented in the field of neuropsychology that executive functions are highly diverse, including but not limited to set-shifting and inhibition abilities (as measured by the Trail Making Test Part B), attention control (as measured by the Symbol Digit Modalities Test), strategic search of the contents of semantic memory and retrieval of appropriate entries (lexical\phonological representations according to specific semantic categories provided by the tester, as assessed by the Semantic Verbal Fluency test), and working memory (as assessed by the digits reverse test). We have added relevant references to support this claim in section 2.3.2.

  1. It is highly recommended to modify Figure 2 and Figure 3, as the images are not of high quality.

Answer: High resolution renderings of figs 2-3 are provided as separate files to address this issue.

Reviewer 4 Report

Thank you for having me review this manuscript. I have a few comments that could improve the quality of the paper. 

1. Avoid using elder or elderly. You can use older adults instead. 

2. It is better to use older adults without dementia than non-demented elderly. Change it throughout the manuscript. 

3. Highlight in the method section that the current study is a secondary analysis study. 

4. Explain how the sample size was calculated. 

5. More information is needed for the Greek version of the Geriatric Depression Scale. 

Author Response

Reviewer 4

Thank you for having me review this manuscript. I have a few comments that could improve the quality of the paper. 

  1. Avoid using elder or elderly. You can use older adults instead. 

Answer: This has now been changed as suggested

  1. It is better to use older adults without dementia than non-demented elderly. Change it throughout the manuscript. 

Answer: This has also been changed as suggested

  1. Highlight in the method section that the current study is a secondary analysis study. 

Answer: We have now highlighted in the methods that this is a secondary analysis.

  1. Explain how the sample size was calculated.

Answer: Given that the study in the context of which the current analyses were conducted was designed to address several complementary research aims (in addition to the ones address in the present work), we estimated the power achieved for multiple regression models with up to 10 IVs (including interaction terms as in the model presented in section 3.2). Using G Power it was estimated that the achieved power was 96% to obtain a significant regression coefficient (at α=0.05, two-tailed) in a model with a total of 10 predictors, assuming even a small effect size of each predictor.

  1. More information is needed for the Greek version of the Geriatric Depression Scale. 

Answer: In the revised manuscript more information about the GDS is now added.

Round 2

Reviewer 4 Report

Thank you for the effort you have put in the revised manuscript.